# CD40 is a major regulator of dendrite growth from developing excitatory and inhibitory neurons

**Paulina Carriba, Alun M Davies***

School of Biosciences, Cardiff University, Cardiff, United Kingdom

**Abstract** Dendrite size and morphology are key determinants of the functional properties of neurons and neural circuits. Here we show that CD40, a member of the TNF receptor superfamily, is a major regulator of dendrite growth and elaboration in the developing brain. The dendrites of hippocampal excitatory neurons were markedly stunted in $Cd40^{-/-}$ mice, whereas those of striatal inhibitory neurons were much more exuberant. These striking and opposite phenotypic changes were also observed in excitatory and inhibitory neurons cultured from $Cd40^{-/-}$ mice and were rescued by soluble CD40. The changes in excitatory and inhibitory neurons cultured from $Cd40^{-/-}$ mice were mimicked in neurons of $Cd40^{+/+}$ mice by treatment with soluble CD40L and were dependent on PKC-β and PKC-γ, respectively. These results suggest that CD40-activated CD40L reverse signalling has striking and opposite effects on the growth and elaboration of dendrites among major classes of brain neurons by PKC-dependent mechanisms.
DOI: https://doi.org/10.7554/eLife.30442.001

## Introduction

The regulation of dendrite growth and elaboration during development has a major bearing on the functional properties of neurons and neural circuits, and many neurodevelopmental and acquired disorders of neural function are due primarily to structural abnormalities of dendrites and their connections (*Penzes et al., 2011*). In addition to intrinsic developmental programmes and the pattern of electrical activity, a wide variety of extrinsic signals orchestrate the growth and remodeling of dendrites during development and maturation, including delta-notch, Eph-Ephrins, cell adhesion molecules, neurotrophins, semaphorins and slits (*Metzger, 2010*; *Valnegri et al., 2015*). One of the latest groups of proteins recognized to modulate growth of neural processes during development is the tumor necrosis factor superfamily (TNFSF), the 19 members of which are best understood for their many roles in the immune system (*Hehlgans and Pfeffer, 2005*). They are active both as membrane-integrated ligands and as soluble ligands following cleavage from the cell membrane, and bind to one or more members of the TNF receptor superfamily (TNFRSF). In addition, several TNFRSF members can also function as ligands for membrane-integrated TNFSF binding partners and initiate reverse signalling (*Sun and Fink, 2007*). Most of the work on the TNFSF and TNFRSF on the regulation of neural process growth has focused on PNS neurons, where several members of these superfamilies act on different kinds of neurons during circumscribed phases of development, either enhancing or inhibiting axon growth (*Desbarats et al., 2003*; *Gavaldà et al., 2009*; *Gutierrez et al., 2013*; *Kisiswa et al., 2013*, *2017*; *McWilliams et al., 2015*; *O'Keeffe et al., 2008*; *Wheeler et al., 2014*). In addition, several studies have implicated TNFSF members in regulating the growth of neural processes in the developing CNS (*Neumann et al., 2002*; *Osório et al., 2014*; *Zuliani et al., 2006*; *McWilliams et al., 2017*).

We have investigated whether CD40 (TNFRSF5) plays a role in regulating the growth and elaboration of dendrites in the developing CNS. CD40 and its ligand CD40L play key roles in immune

*For correspondence:
DaviesAlun@cardiff.ac.uk

responses and in the pathogenesis of autoimmune disease (*Peters et al., 2009*). CD40 and CD40L have also recently been implicated in regulating sympathetic axon growth and are required for establishing the innervation of particular tissues in the developing PNS (*McWilliams et al., 2015*). Here, we focused on the two most extensively characterized populations of excitatory and inhibitory projection neurons in the mouse brain, the excitatory glutamatergic pyramidal neurons of the hippocampus and the inhibitory GABA-ergic medium spiny neurons (MSNs) of the striatum. Hippocampal pyramidal neurons are one of the best-characterized models for studying the differentiation and growth of axons and dendrites during development (*Kaech and Banker, 2006*). MSNs comprise the majority of neurons in the striatum, have extensive dendritic arbors, project to the surrounding nuclei of the basal ganglia and their degeneration is the pathognomonic feature of Huntington's disease (*Reiner et al., 1988*). Both kinds of neurons are easily recognized by their distinctive morphologies in vivo and can be cultured at a stage when they are elaborating dendrites and are the major cell type in culture. Our demonstration that CD40-activated CD40L reverse signaling has pronounced and distinctive effects on the growth and elaboration of dendrites from different classes of neurons in vitro and in vivo has important implications for neural development and the establishment and function of neural circuits.

## Results

### Phenotypic changes in the dendrites of excitatory and inhibitory neurons of *Cd40*[-/-] mice

Golgi preparations carried out on *Cd40*[+/+] and *Cd40*[-/-] mice at postnatal day 10 (P10) revealed that the size and complexity of the dendritic arbors of hippocampal pyramidal neurons and striatal MSNs were markedly affected in the absence of CD40, but in opposite ways. The dendritic arbors of the excitatory pyramidal neurons of the CA fields of the hippocampus of *Cd40*[-/-] mice were dramatically stunted compared with those of *Cd40*[+/+] littermates (*Figure 1A*). Representative high power images show that the reduction of dendrite size and complexity in *Cd40*[-/-] mice affected both apical and basal dendrite compartments of these excitatory neurons (*Figure 1B*). Because the dendrite arbors of pyramidal neurons in CA1 are normally less exuberant than those of other CA regions, CA1 pyramidal dendrites were especially amenable to analysis, which was carried out separately on the apical and basal compartments. This revealed marked and highly significant reductions in total dendrite length in both apical (p=$6.02\times10^{-16}$) and basal (p=$4.75\times10^{-14}$) compartments of *Cd40*[-/-] mice compared with *Cd40*[+/+] littermates (*Figure 1C*). These reductions were in reflected in the Sholl analyses (*Figure 1D*), which plot branching in the dendrite arbors with distance from the cell body (two-way ANOVA *Cd40*[+/+] versus *Cd40*[-/-], p=$1.67\times10^{-14}$, apical dendrites and p=$4.75\times10^{-5}$, basal dendrites). Accordingly, quantification of the number of branch points in these dendrite compartments showed clear statistically significant reductions in *Cd40*[-/-] mice (p=$8.85\times10^{-7}$, apical and p=$6.48\times10^{-8}$, basal), however there was no significant difference (p=0.82) in the number of primary dendrites between the pyramidal neurons of *Cd40*[-/-] and *Cd40*[+/+] mice (*Figure 1—figure supplement 1*). Taken together, these findings indicate that CD40 plays a major role in promoting the growth and elaboration of hippocampal pyramidal neuron dendrites.

In marked contrast to hippocampal excitatory neurons, the dendritic arbors of the striatal MSNs of *Cd40*[-/-] mice were very much larger and more profuse than those of *Cd40*[+/+] littermates (*Figure 1E and F*). Quantification of total dendrite length revealed a marked and highly significant increase in *Cd40*[-/-] mice compared with *Cd40*[+/+] littermates (p=$6.87\times10^{-17}$) (*Figure 1G*). This increase in dendrite size was in reflected in the Sholl analysis (*Figure 1H*) (two-way ANOVA *Cd40*[+/+] versus *Cd40*[-/-], p=$6.68\times10^{-4}$). Quantification of the number of branch points in these dendrite compartments showed clear statistically significant increase in *Cd40*[-/-] mice (49% increase, p=$1.13\times10^{-13}$), and as with hippocampal pyramidal neurons, there was no significant difference in the number of primary dendrites in the MSNs of *Cd40*[-/-] and *Cd40*[+/+] mice (*Figure 1—figure supplement 1*). These findings indicate that CD40 plays a major role in restricting the growth and elaboration of MSN dendrites in the developing striatum.

To ascertain whether the striking phenotypic changes observed in the hippocampus and striatum of early postnatal *Cd40*[-/-] mice are maintained in older mice, we compared *Cd40*[-/-] and *Cd40*[+/+] littermates in P30 and adult mice. Golgi preparations revealed that hippocampal pyramidal neuron

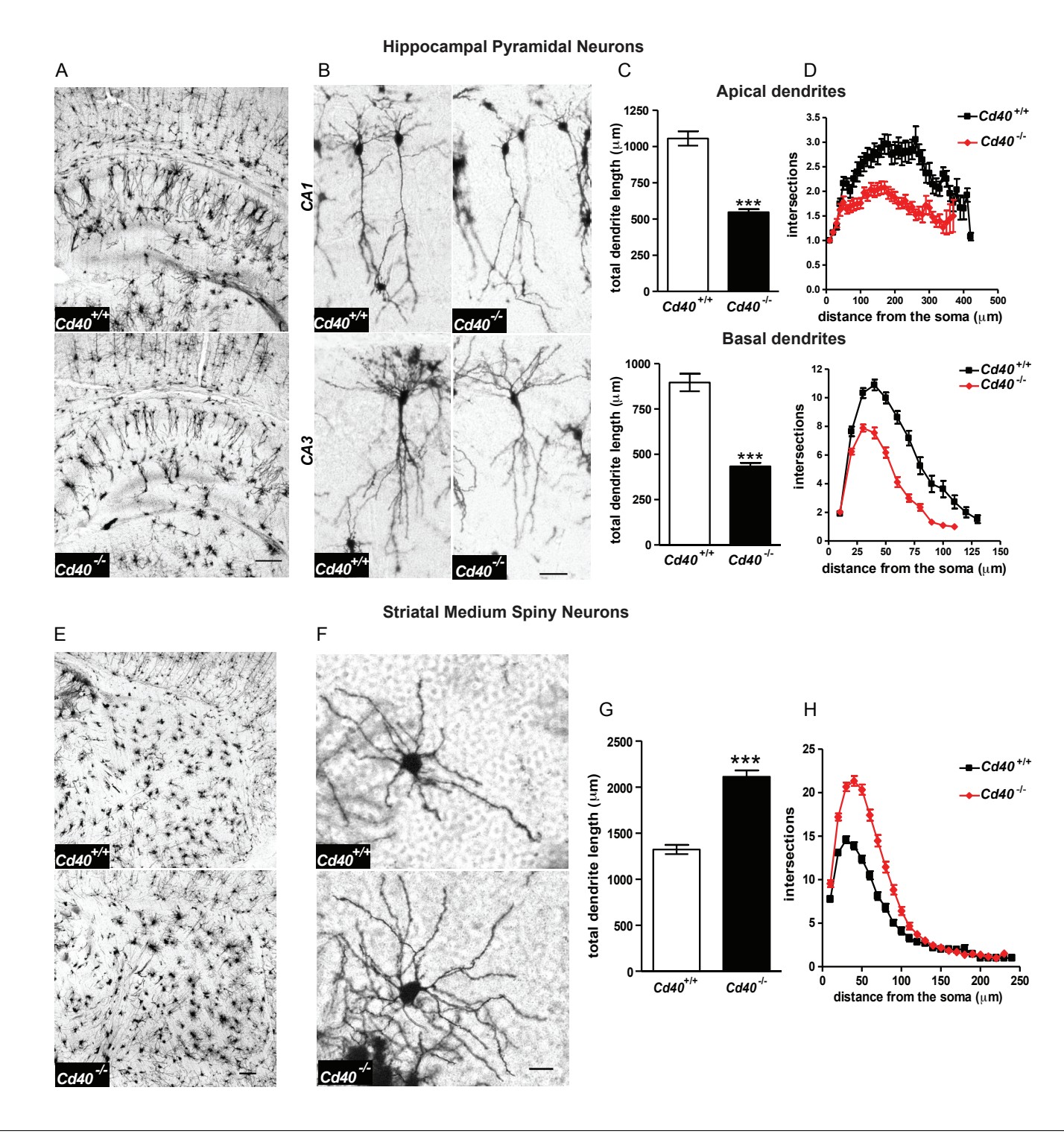

**Figure 1.** Phenotypic changes in hippocampal pyramidal neurons and striatal MSNs in P10 *Cd40*[-/-] mice. (**A**) Representative low-power images of Golgi preparations of the hippocampal CA regions and dentate gyrus of *Cd40*[+/+] and *Cd40*[-/-] mice. Scale bar, 250 μm. (**B**) High-power images of CA1 and CA3 pyramidal neurons of *Cd40*[+/+] and *Cd40*[-/-] mice. Scale bar, 50 μm. (**C**) Quantification of the total apical dendrite lengths (*Cd40*[+/+], n = 74 and *Cd40*[-/-], n = 67) and total basal dendrite lengths (*Cd40*[+/+], n = 51 and *Cd40*[-/-], n = 49) of CA1 pyramidal neurons of *Cd40*[+/+] and *Cd40*[-/-] mice. (**D**) Sholl plots of the apical (*Cd40*[+/+], n = 86 and *Cd40*[-/-], n = 71) and basal (*Cd40*[+/+], n = 64 and *Cd40*[-/-], n = 53) dendrites of CA1 pyramidal neurons. (**E**) Representative low-power images of Golgi preparations of the striatum of *Cd40*[+/+] and *Cd40*[-/-] mice. Scale bar, 250 μm. (**F**) Representative high-power images of MSNs of *Cd40*[+/+] and *Cd40*[-/-] mice. Scale bar, 20 μm. (**G**) Quantification of the total dendrite lengths of MSNs (*Cd40*[+/+], n = 85 and *Cd40*[-/-],
*Figure 1 continued on next page*

Figure 1 continued

n = 89). (H) Sholl plots of MSN dendrites ($Cd40^{+/+}$, n = 85 and $Cd40^{-/-}$, n = 89). The means ± s.e.m are shown in C, D, G and H. Data were obtained from neurons analysed in at least three mice of each genotype, ***p<0.0001 t-test (actual p values provided in the text). Sholl plots were analysed by two-way ANOVA with Bonferroni post tests. The statistical significances for each distance from the soma are included in *Figure 1—source data 3*.
DOI: https://doi.org/10.7554/eLife.30442.002

The following source data and figure supplements are available for figure 1:

Source data 1. Data of individual Golgi-stained pyramidal neurons in $Cd40^{+/+}$ and $Cd40^{-/-}$ mice.
DOI: https://doi.org/10.7554/eLife.30442.005

Source data 2. Data of individual Golgi-stained medium spiny neurons in $Cd40^{+/+}$ and $Cd40^{-/-}$ mice.
DOI: https://doi.org/10.7554/eLife.30442.006

Source data 3. 2-way ANOVA for pyramidal neurons and medium spiny neurons in $Cd40^{+/+}$ and $Cd40^{-/-}$ mice.
DOI: https://doi.org/10.7554/eLife.30442.007

Figure supplement 1. Opposite effects of the Cd40 null mutation on dendrite branching in hippocampal pyramidal neurons and striatal MSNs.
DOI: https://doi.org/10.7554/eLife.30442.003

Figure supplement 2. Phenotypic changes are maintained in P30 and adult $Cd40^{-/-}$ mice.
DOI: https://doi.org/10.7554/eLife.30442.004

dendrite arbors were clearly stunted in $Cd40^{-/-}$ mice compared with $Cd40^{+/+}$ littermates whereas striatal MSN dendrite arbors of $Cd40^{-/-}$ mice were much larger and more complex than those of $Cd40^{+/+}$ littermates both ages (*Figure 1—figure supplement 2*). This shows that the distinctive phenotypic changes evident in the dendrite arbors of both kinds neurons observed in the early postnatal period are maintained throughout life.

## Dendrite growth and elaboration is regulated by CD40-activated CD40L reverse signalling

Interaction of CD40 and membrane-integrated CD40L can initiate bi-directional signaling: CD40L-activated CD40-mediated forward signaling and CD40-activated CD40L-mediated reverse signaling (*Sun and Fink, 2007*). To determine which signaling mechanism is responsible for the effects of CD40 on dendrite growth and elaboration in hippocampal pyramidal neurons and striatal MSNs, we established cultures of these neurons from $Cd40^{-/-}$ and $Cd40^{+/+}$ mice and studied the effect of soluble CD40L and soluble CD40.

Hippocampal pyramidal neuron cultures were established from E18 mice. By 9 days in vitro, MAP-2-positive dendrites can be unambiguously identified and quantified (*Kaech and Banker, 2006*; *Osório et al., 2014*). Similar to the phenotypic changes observed in Golgi preparations of $Cd40^{-/-}$ mice, pyramidal neurons cultured from $Cd40^{-/-}$ mice were smaller than those cultured from $Cd40^{+/+}$ mice (*Figure 2A*). Measurement of the total dendrite length in the dendrite arbors of individual neurons revealed a highly significant reduction in cultures established from $Cd40^{-/-}$ mice compared with those cultured from $Cd40^{+/+}$ mice ($p=2.9\times10^{-6}$) (*Figure 2B*), but no significant difference in the number of primary dendrites (*Figure 2—figure supplement 1*). These data suggest that the morphologies of hippocampal pyramidal neurons cultured from $Cd40^{+/+}$ and $Cd40^{-/-}$ mice reflect the phenotypic differences observed in vivo.

Treatment of pyramidal neurons cultured from $Cd40^{+/+}$ mice with soluble CD40L significantly reduced dendrite length to the same extent as that observed in untreated pyramidal neurons cultured from $Cd40^{-/-}$ mice ($p=1.1\times10^{-8}$) (*Figure 2B*). Neither soluble CD40 (CD40-Fc chimera, in which the extracellular domain of CD40 is linked to the Fc part of human IgG1) nor control Fc protein had any significant effect on dendrite growth from pyramidal neurons cultured from $Cd40^{+/+}$ mice. While CD40L reduced dendrite growth from wild type neurons, it did not significantly affect dendrite growth from pyramidal neurons of $Cd40^{-/-}$ mice (*Figure 2B*), suggesting that CD40L does not exert a non-specific suppressive effect on dendrite growth. The most parsimonious explanation for these results is that addition of CD40L to pyramidal neurons cultured from $Cd40^{+/+}$ mice competes with endogenous membrane integrated CD40L for binding to endogenous CD40, thereby blocking CD40-activated CD40L-mediated reverse signalling. Further support for CD40-activated CD40L-mediated reverse signaling came from phenotype rescue experiments. CD40-Fc, but not Fc control protein, fully restored the reduced dendrite growth phenotype of pyramidal neurons cultured from $Cd40^{-/-}$ mice ($p=6.5\times10^{-3}$) (*Figure 2B*). Taken together, these observations suggest that CD40-

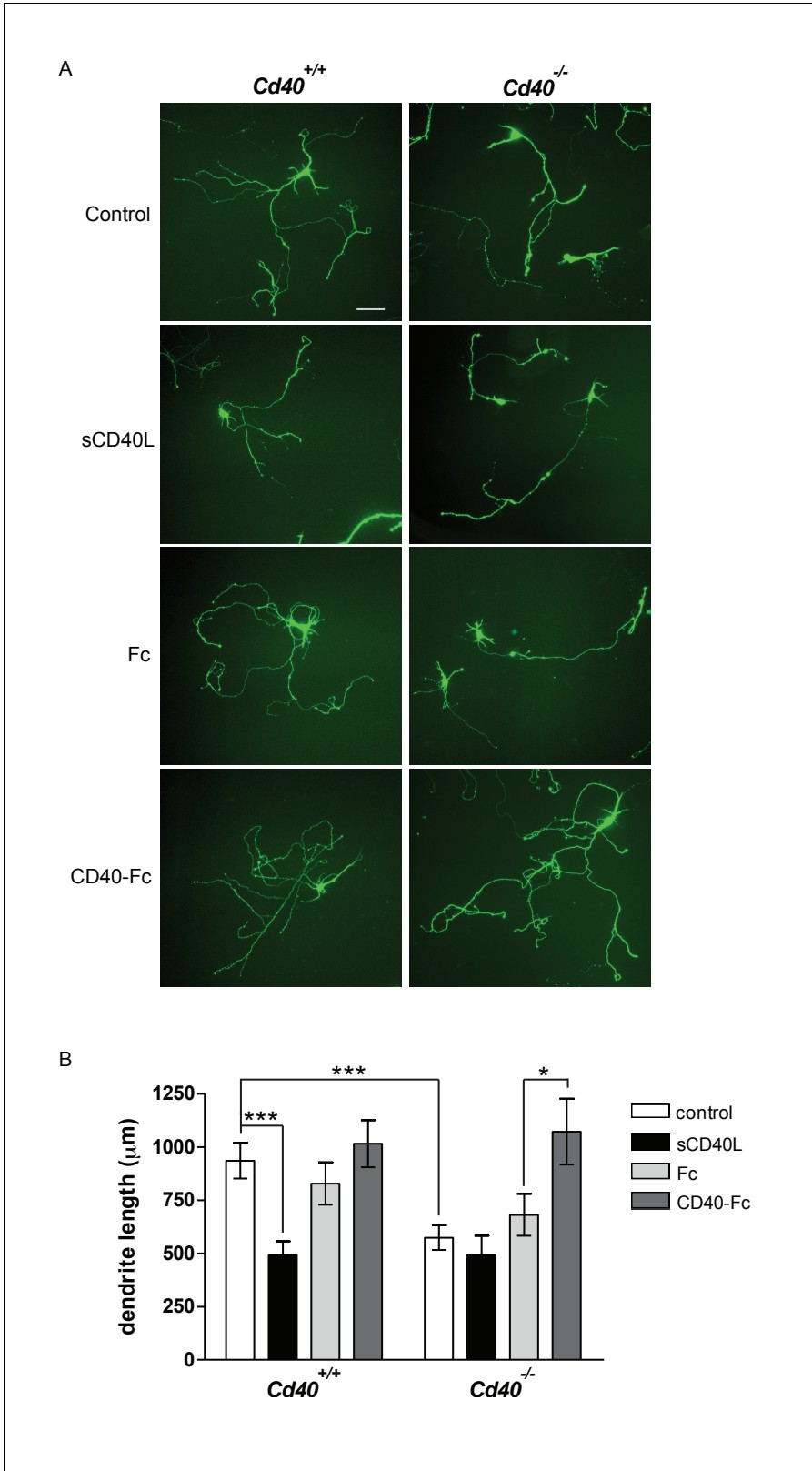

**Figure 2.** CD40-activated CD40L reverse signalling enhances dendrite growth from cultured hippocampal pyramidal neurons. (**A**) Representative photomicrographs of hippocampal neurons of $Cd40^{+/+}$ and $Cd40^{-/-}$ E18 embryos cultured for 9 days and treated 24 hr after plating with either soluble CD40L (sCD40L), CD40-Fc or Fc protein (each at 1 µg/ml) or untreated (control). Scale bar, 100 µm. (**B**) Quantification of total dendrite lengths of neurons in these cultures after 9 days in vitro. Mean ± s.e.m of data collected from three independent experiments from $Cd40^{+/+}$ (control, n = 70;
*Figure 2 continued on next page*

*Figure 2 continued*

sCD40L, n = 51; Fc, n = 55; CD40-Fc, n = 61) and *Cd40*<sup>-/-</sup> (control, n = 60; sCD40L, n = 57; Fc, n = 61; CD40-Fc, n = 59). One-way ANOVA with multiple Newman-Keuls statistical comparison with control, *p<0.01, and ***p<0.0001 (actual p values provided in the text).

DOI: https://doi.org/10.7554/eLife.30442.008

The following source data and figure supplement are available for figure 2:

**Source data 1.** Means, s.e.m. and n numbers for bar charts.

DOI: https://doi.org/10.7554/eLife.30442.010

**Figure supplement 1.** Number of primary dendrites per neuron in the neurite arbors of cultured hippocampal pyramidal neurons.

DOI: https://doi.org/10.7554/eLife.30442.009

activated CD40L mediated reverse signalling promotes dendrite growth from developing hippocampal pyramidal neurons.

In addition to dendrites, axons are clearly identifiable in cultures of developing hippocampal pyramidal neurons (*Dotti et al., 1988*; *Kaech and Banker, 2006*). Axons are the first processes to emerge from the cell body and remain the longest process in longer-term cultures. This permitted us to investigate the potential influence of CD40 signalling on pyramidal axon growth (*Figure 3*). The axons of pyramidal neurons cultured from *Cd40*<sup>-/-</sup> mice were significantly shorter than the axons of neurons cultured from *Cd40*<sup>+/+</sup> mice in both 3 day and 9 day cultures (p=$7.8\times10^{-12}$, 3 days; p=$1.3\times10^{-4}$, 9 days). The effects of CD40L and CD40-Fc on axon growth were very similar to the effects of these reagents on dendrite growth. CD40L, but not CD40-Fc, suppressed axon growth from *Cd40*<sup>+/+</sup> neurons to the same extend as that observed in untreated *Cd40*<sup>-/-</sup> neurons (p=$6.4\times10^{-11}$, 3 days; p=$1.2\times10^{-5}$, 9 days). The reduced axon growth phenotype of neurons of *Cd40*<sup>-/-</sup> mice was completely rescued by CD40-Fc (p=$2.6\times10^{-5}$, 3 days; p=$4.9\times10^{-6}$, 9 days). These results suggest that CD40-activated CD40L-mediated reverse signalling also enhances axon growth from developing hippocampal pyramidal neurons in culture. However, because the full extent of these axons cannot be reliably discerned in Golgi preparations, we cannot definitively conclude that CD40L reverse signalling enhances the growth of pyramidal axons in vivo.

MSN cultures were established from E14 striatal primordia. Although MSNs comprise the vast majority of neurons in the striatum, some contamination by adjoining regions is inevitable when dissecting striatal primordia from the E14 brain. For this reason, we positively identified MSNs using an antibody to dopamine and cyclic AMP-regulated protein (DARPP-32), which is expressed by more than 95% of MSNs (*Ouimet et al., 1998*). Similar to the phenotypic changes observed in Golgi preparations of *Cd40*<sup>-/-</sup> mice, the neurite arbors of MSNs cultured from *Cd40*<sup>-/-</sup> mice were larger than those of MSNs cultured from *Cd40*<sup>+/+</sup> mice (*Figure 4A*). Although dendrites and axons cannot be easily distinguished in these cultures, measurement of total neurite length confirmed highly significant increases in neurons cultured from *Cd40*<sup>-/-</sup> mice compared with those cultured from *Cd40*<sup>+/+</sup> littermates (p=$7.2\times10^{-10}$) (*Figure 4B*). This suggests, as with hippocampal pyramidal neurons, that the morphologies of MSNs cultured from *Cd40*<sup>+/+</sup> and *Cd40*<sup>-/-</sup> mice reflect the phenotypic differences observed in vivo.

Treatment of MSNs cultured from *Cd40*<sup>+/+</sup> mice with soluble CD40L caused a significant increase in total neurite length (p=$1.57\times10^{-6}$) that was similar to the total neurite length observed in untreated MSNs cultured from *Cd40*<sup>-/-</sup> mice (*Figure 4B*). CD40-Fc did not affect neurite growth from MSNs cultured from *Cd40*<sup>+/+</sup> mice, but significantly reduced the more exuberant neurite growth of MSNs cultured from *Cd40*<sup>-/-</sup> mice (p=$3.6\times10^{-14}$) to the level observed in untreated MSNs cultured from *Cd40*<sup>+/+</sup> mice (*Figure 4B*). These observations suggest that CD40 normally restricts the growth of neural processes from developing MSNs by a CD40L-mediated reverse signalling mechanism.

## PKC mediates both the enhancing and inhibitory effects of CD40-activated CD40L reverse signalling on Dendrite growth

Because we have shown that activation of protein kinase C (PKC) is an essential step in the axon growth response of developing sympathetic neurons to TNFR1-activated TNFα-mediated reverse signaling (*Kisiswa et al., 2017*), we investigated whether PKC plays a role in mediating the divergent effects CD40-activated CD40L reverse signalling on dendrite growth in excitatory and/or inhibitory neurons. PKC is a family of at least 12 serine/threonine kinases that play key roles in several

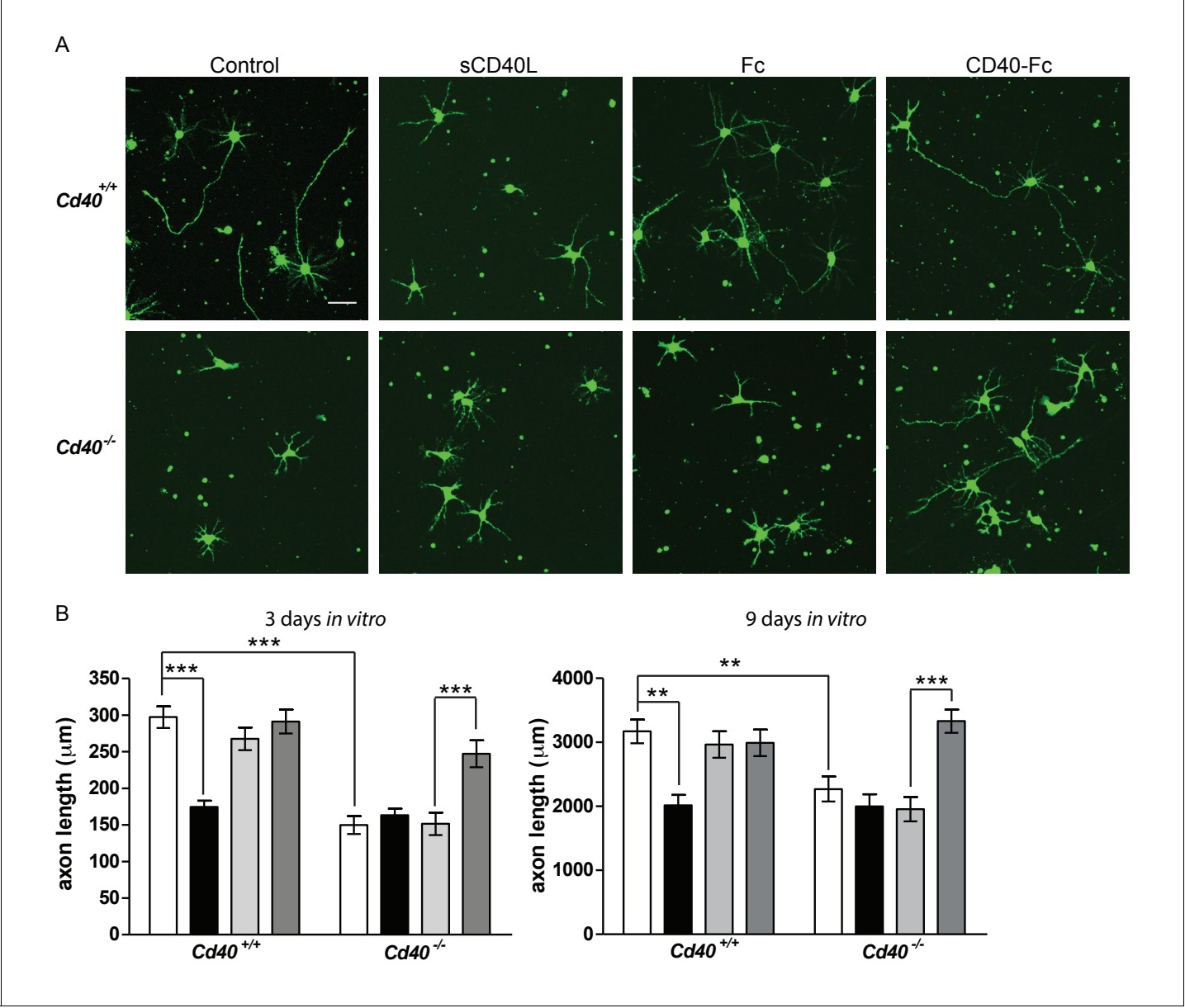

**Figure 3.** CD40-activated CD40L reverse signalling enhances axon growth from cultured hippocampal pyramidal neurons. (**A**) Representative photomicrographs of neurons of *Cd40*[+/+] and *Cd40*[-/-] E18 embryos cultured for 3 days and treated 24 hr after plating with either soluble CD40L (sCD40L), CD40-Fc or Fc protein (each at 1 μg/ml) or untreated (control). Scale bar, 50 μm. (**B**) Quantification of axon length from hippocampal pyramidal neurons of *Cd40*[+/+] and *Cd40*[-/-] E18 embryos cultured for 3 and 9 days. Mean ± s.e.m of data collected from at least three independent experiments at 3 days from *Cd40*[+/+] (control, n = 74; sCD40L, n = 68; Fc, n = 72; CD40-Fc, n = 77) and *Cd40*[-/-] (control, n = 61; sCD40L, n = 64; Fc, n = 56; CD40-Fc, n = 81); and at 9 days from *Cd40*[+/+] (control, n = 70; sCD40L, n = 51; Fc, n = 55; CD40-Fc, n = 61) and *Cd40*[-/-] (control, n = 60; sCD40L, n = 57; Fc, n = 61; CD40-Fc, n = 59). One-way ANOVA with multiple Newman-Keuls statistical comparisons test between the conditions indicated, **$p < 0.001$ and ***$p < 0.0001$ (actual p values provided in the text).

DOI: https://doi.org/10.7554/eLife.30442.011

The following source data is available for figure 3:

**Source data 1.** Means, s.e.m. and n numbers for bar charts.
DOI: https://doi.org/10.7554/eLife.30442.012

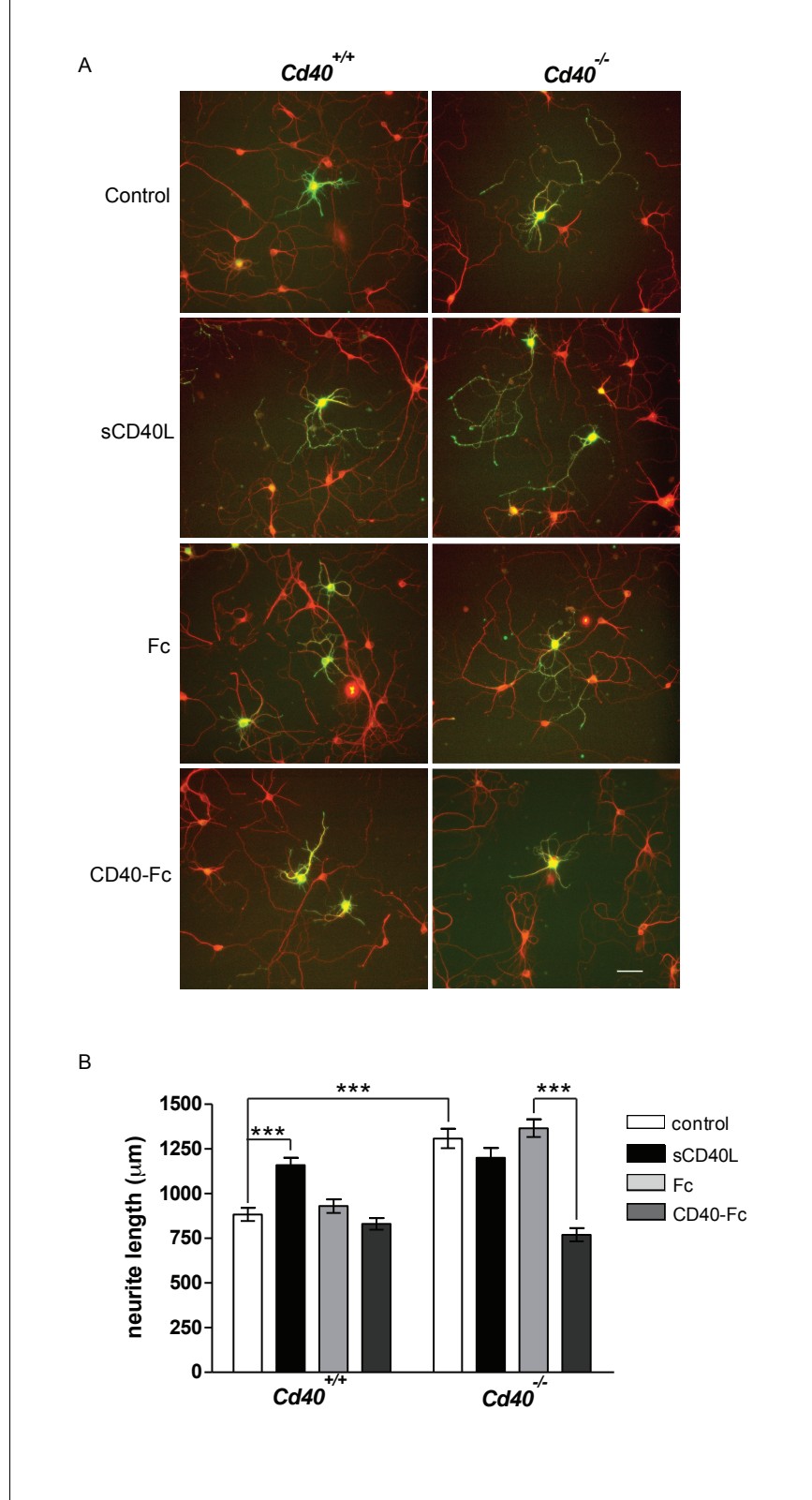

**Figure 4.** CD40-activated CD40L reverse signalling reduces neurite growth from cultured MSNs. (**A**) Representative photomicrographs of MSNs of Cd40+/+ and Cd40-/- E14 embryos cultured for 10 days and treated with either soluble CD40L (sCD40L) at 0.5 μg/ml, CD40-Fc or Fc protein (each at 1 μg/ml) or untreated (control). Neurons were double labelled for βIII tubulin (red) and DARPP-32 (green) to identify MSNs. Scale bar, 50 μm. (**B**) *Figure 4 continued on next page*

*Figure 4 continued*

Quantification of total neurite lengths of MSNs after 10 days in vitro. Mean ± s.e.m of data collected from three independent experiments from *Cd40⁺/⁺* (control, n = 69; sCD40L, n = 65; Fc, n = 63; CD40-Fc, n = 62) and *Cd40⁻/⁻* (control, n = 60; sCD40L, n = 62; Fc, n = 52; CD40-Fc, n = 73). One-way ANOVA with multiple Newman-Keuls statistical comparisons with control. ***p<0.0001 (actual p values provided in the text).
DOI: https://doi.org/10.7554/eLife.30442.013

The following source data is available for figure 4:

**Source data 1.** Means, s.e.m. and n numbers for bar charts.
DOI: https://doi.org/10.7554/eLife.30442.014

signaling pathways that regulate a diversity of cellular functions (*Mellor and Parker, 1998*). In these studies, we asked whether pharmacological manipulation of PKC activity in CD40-deficient neurons could inhibit or mimic the effect of CD40-Fc on dendrite growth on hippocampal pyramidal neurons and striatal MSNs cultured from *Cd40⁻/⁻* mice. PCK activity was inhibited by the pan-PKC inhibitor Go6983 (*Gschwendt et al., 1996*) and PKC activation was mimicked by phorbol-12-myristate 13-acetate (PMA), an analogue of diacylglycerol which participates in the activation of conventional PKCs (PKCα, PKCβ and PKCγ) and novel PKCs (PKCδ, PKCε, PKCθ and PKCη) (*Liu and Heckman, 1998*).

In cultures of hippocampal pyramidal neurons established from E18 *Cd40⁻/⁻* mice, the significant restoration of dendrite growth by CD40-Fc (p=3.8×10⁻⁴, Fc treatment compared with CD40-Fc treatment) was completely prevented by Go6983 (p=1.6×10⁻⁵, CD40-Fc treatment compared with CD40-Fc plus Go6983 treatment), although Go6983 had no effect on dendrite growth on its own (*Figure 5A and B*). PMA significantly rescued the stunted dendrite phenotype of hippocampal pyramidal neurons established from *Cd40⁻/⁻* mice (p=1.6×10⁻²) almost as effectively as CD40-Fc (*Figure 5A and B*). DMSO, the vehicle for Go6983 and PMA, had no significant effect on dendrite growth (not shown). Neither Go6983 nor PMA had any significant effect on the number of pyramidal neuron primary dendrites (not shown). These results suggest that PKC activation mediates the effect of CD40-activated CD40L-mediated reverse signalling on dendrite growth from developing hippocampal pyramidal neurons.

In cultures of striatal MSNs established from E14 *Cd40⁻/⁻* mice, the significant suppression of dendrite growth by CD40-Fc (p=9.3×10⁻¹³, Fc treatment compared with CD40-Fc treatment) was completely inhibited by Go6983 (p=2.5×10⁻⁷, CD40-Fc treatment compared with CD40-Fc plus Go6983 treatment) and as in hippocampal neuron cultures, Go6983 had no effect on MSN dendrite growth on its own (*Figure 5C and D*). PMA significantly suppressed dendrite growth from MSNs (p=4.5×10⁻¹⁷) to the same extent as CD40-Fc. Neither Go6983 nor PMA had any significant effect on the number of MSN primary dendrites (not shown). These results suggest that PKC activation mediates the effect of CD40-activated CD40L on dendrite growth suppression from developing MSNs.

## PKC-β and PKC-γ mediate the opposite effects of CD40-activated CD40L reverse signalling on hippocampal and striatal neurons, respectively

To investigate whether particular PKC isoforms selectively participate in mediating the effects of CD40-activated CD40L reverse signalling on dendrite growth, we studied the effects of siRNA knock down of selected PKC isoforms. Because PMA strongly activates conventional and novel PKC isoforms (*Wu-Zhang and Newton, 2013*), we first checked expression of the conventional and novel PKC isoforms that have been reported to be abundantly expressed in whole brain. Using western blotting, we showed abundant expression of PKC-α, β, γ and ε in both the hippocampus and striatum (*Figure 6—figure supplement 1*). We subsequently transfected cultures of hippocampal and striatal neurons with siRNA oligonucleotides directed to each of these isoforms and with a control RNA that has a scrambled sequence, and used western blotting to ascertain whether knockdown was successful. Because we were able to consistently and efficiently knock down PKC-β and PKC-γ expression with specific siRNA in these cultures (*Figure 6—figure supplement 1*), we focused subsequent experiments on these isoforms.

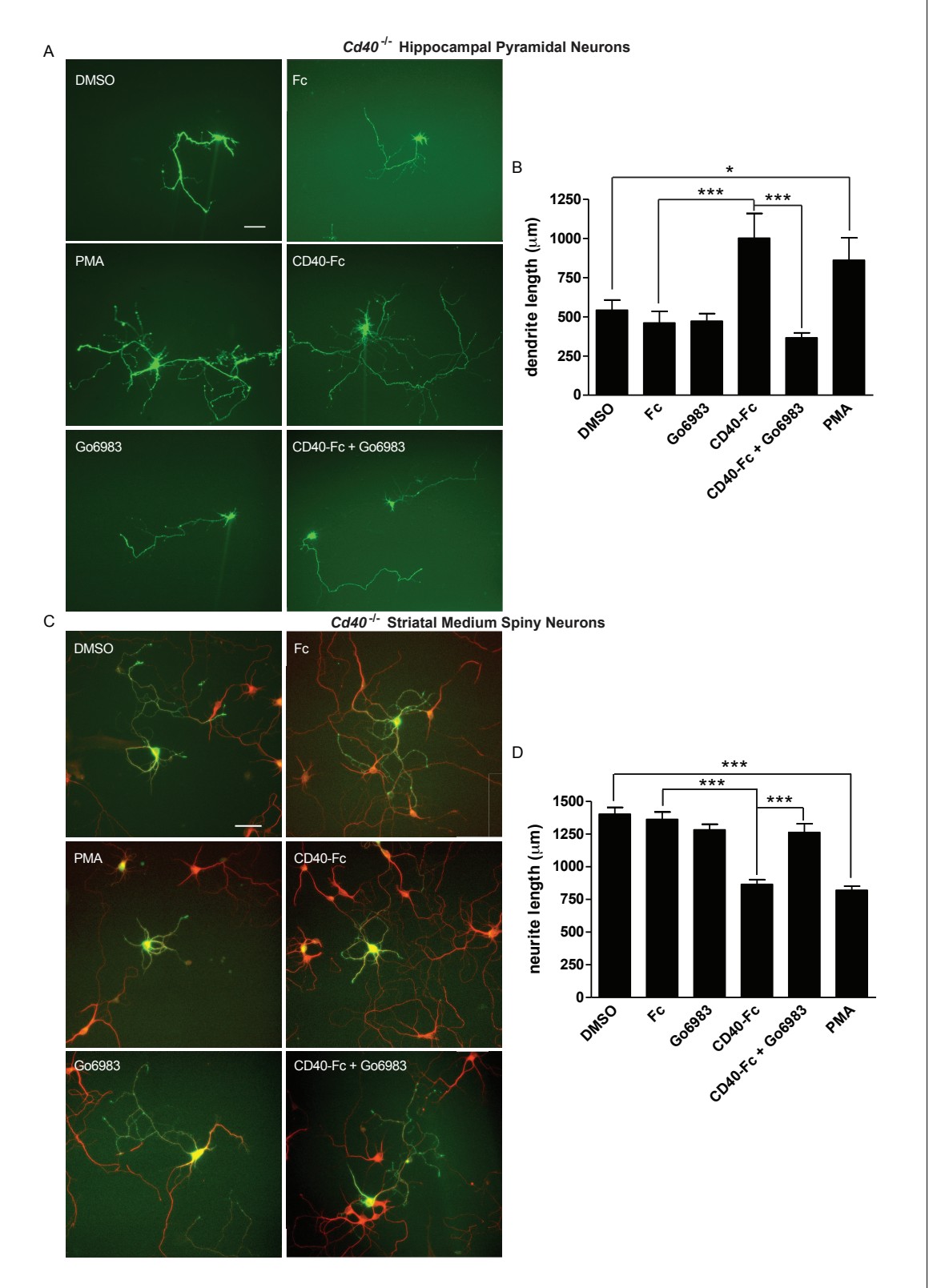

**Figure 5.** PKC mediates the effect of CD40-activated CD40L reverse signalling on dendrite growth. (**A**) Representative photomicrographs of hippocampal neurons of *Cd40⁻/⁻* E18 embryos cultured for 9 days and treated 24 hr after plating with either 1 μg/ml CD40-Fc, 1 μg/ml Fc protein, 500 nM Go6983, 500 nM PMA or the equivalent level of vehicle DMSO. Scale bar, 100 μm. (**B**) Quantification of total dendrite lengths of neurons in these cultures after 9 days in vitro. Mean ± s.e.m of data collected from three independent experiments in *Cd40⁻/⁻* (control, n = 43; DMSO, n = 43; Go6983,

*Figure 5 continued on next page*

*Figure 5 continued*

n = 34; Fc, n = 33; CD40-Fc, n = 37; CD40-Fc + Go6983, n = 37; PMA, n = 34). (**C**). Representative photomicrographs of MSNs of *Cd40$^{-/-}$* E14 embryos cultured for 10 days and treated as indicated. MSNs were identified by double labelling for βIII tubulin (red) and DARPP-32 (green). Scale bar, 50 μm. (**D**) Quantification of total neurite lengths of MSNs after 10 days in vitro. Mean ± s.e.m of data collected from three independent experiments in *Cd40$^{-/-}$* (control, n = 56; DMSO, n = 46; Go6983, n = 49; Fc, n = 50; CD40-Fc, n = 61; CD40-Fc + Go6983, n = 48; PMA, n = 58). *p<0.01 and ***p<0.0001, statistical comparison with the equivalent control (Fc for CD40-Fc and DMSO for PMA, actual p values provided in the text), one-way ANOVA with multiple Newman-Keuls tests.

DOI: https://doi.org/10.7554/eLife.30442.015

The following source data is available for figure 5:

**Source data 1.** Means, s.e.m. and n numbers for pyramidal neuron bar charts.
DOI: https://doi.org/10.7554/eLife.30442.016
**Source data 2.** Means, s.e.m. and n numbers for medium spiny neuron bar charts.
DOI: https://doi.org/10.7554/eLife.30442.017

Initial siRNA experiments were carried out on neurons cultured from *Cd40$^{+/+}$* mice. Compared with dendrite growth from hippocampal pyramidal neurons transfected with control siRNA, dendrite growth was significantly decreased by PKC-β siRNA (p=4.9×10$^{-3}$) but not by PKC-γ siRNA (*Figure 6A and B*). Conversely, PKC-γ siRNA but not PKC-β siRNA significantly increased dendrite growth from MSNs compared with control siRNA (p=1.6×10$^{-4}$) (*Figure 6C and D*). This suggests that PKC-β and PKC-γ participate in regulating dendrite growth from developing hippocampal pyramidal neurons and MSNs, respectively.

We subsequently tested whether these siRNAs selectively inhibited the rescue of the dendrite growth phenotype of neurons cultured *Cd40$^{-/-}$* mice by CD40-Fc. Whereas CD40-Fc rescued the impaired dendrite growth of CD40-deficient hippocampal pyramidal neurons transfected with control siRNA, PKC-β siRNA completely prevented CD40-Fc rescue (p=5.7×10$^{-8}$) (*Figure 6A and B*). Whereas CD40-Fc reduced the exuberant dendrite growth of CD40-deficient MSNs transfected with control siRNA, PKC-γ siRNA significantly impaired this effect of CD40-Fc (p=7.4×10$^{-4}$) (*Figure 6C and D*). Taken together, these results implicate PKC-β and PKC-γ in mediating the effects of CD40L reverse signalling in hippocampal pyramidal neurons and MSNs, respectively.

## Expression of CD40 and CD40L

The striking and opposite effects of CD40-activated CD40L-mediated reverse signalling on the growth and elaboration of the dendritic arbors of excitatory hippocampal pyramidal neurons and inhibitory striatal MSNs raised the question of the cellular source of CD40 and how it activates CD40L in vivo. To address this question, we used western blotting and immunochemistry to ascertain the timing of CD40 and CD40L expression in the hippocampus and striatum. Western blotting revealed that both CD40 and CD40L are expressed in the hippocampus and striatum throughout development and in the adult, with highest levels of both proteins postnatally (*Figure 7—figure supplement 1*). CD40 immunoreactivity was lost in tissue obtained from *Cd40$^{-/-}$* mice (*Figure 7—figure supplement 1A*), confirming antibody specificity. Immunohistochemisty revealed that CD40 and CD40L are widely expressed in the hippocampus (*Figure 7*) and striatum (*Figure 8*). Double labelling with anti-MAP2, which labels dendrites, showed that CD40 and CD40L are expressed throughout the dendrite fields of pyramidal and striatal neurons and double labelling with anti-GFAP, which labels astrocytes, showed that CD40 and CD40L are also expressed by these cells. CD40 immunoreactivity was not observed in hippocampal and striatal sections of *Cd40$^{-/-}$* mice and no labelling was observed when the primary antibody was omitted (*Figures 7* and *8*). Double labelling for CD40 and CD40L in dissociated hippocampal and striatal cultures suggest that the great majority of pyramidal neurons and MSNs co-express CD40 and CD40L (*Figure 9*).

## Discussion

We have discovered that CD40 is a major novel regulator of dendrite arborisation in the developing brain that exerts prominent disparate effects on major populations of excitatory and inhibitory neurons. Analysis of Golgi preparations of CD40-deficient postnatal mice revealed that the apical and basal dendrite arbors of excitatory hippocampal pyramidal neurons were very much smaller and less

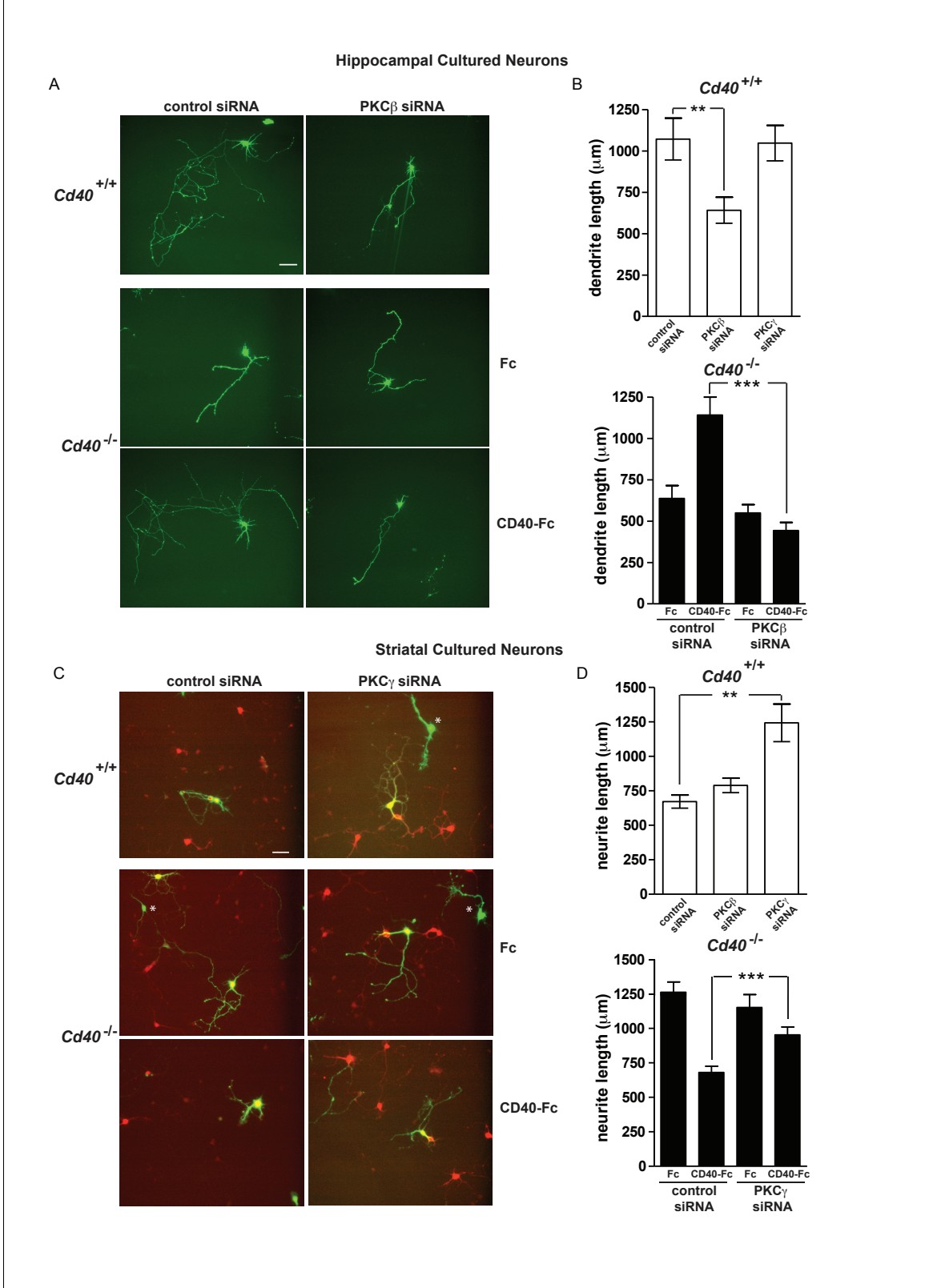

**Figure 6.** PKC-β and PKC-γ selectively participate in mediating the effects of CD40-activated CD40L reverse signalling on dendritic growth. (A) Representative photomicrographs of hippocampal neurons of *Cd40*$^{+/+}$ and *Cd40*$^{-/-}$ E18 embryos transfected with either control siRNA or PKC-β siRNA and cultured for 9 days. The *Cd40*$^{-/-}$ neurons were treated with either 1 μg/ml Fc protein or 1 μg/ml CD40-Fc. Scale bar, 100 μm. (B) Quantification of dendrite length from hippocampal pyramidal neurons of *Cd40*$^{+/+}$ and *Cd40*$^{-/-}$ E18 embryos cultured for 9 days. Mean ± s.e.m of data collected from

*Figure 6 continued on next page*

*Figure 6 continued*

four separate experiments of *Cd40*^+/+^ neurons (control siRNA transfected, n = 32; PKC-β siRNA transfected, n = 33; PKC-γ siRNA transfected, n = 27) and *Cd40*^-/-^ neurons (control siRNA transfected, Fc treated, n = 39; control siRNA transfected, CD40-Fc treated, n = 35; PKC-β siRNA transfected, Fc treated, n = 39; PKC-β siRNA transfected, CD40-Fc treated, n = 39). (C) Representative photomicrographs of MSNs of *Cd40*^+/+^ and *Cd40*^-/-^ E14 embryos transfected with either control siRNA or PKC-γ siRNA and cultured for 10 days. The *Cd40*^-/-^ neurons were treated with either 1 μg/ml Fc protein or 1 μg/ml CD40-Fc. MSNs were identified by DARPP-32 (red) and transfected neurons by the expression of GFP (green). * indicates transfected neurons that were not positive for DARPP-32. Scale bar, 50 μm. (D) Quantification of neurite length from MSNs of *Cd40*^+/+^ and *Cd40*^-/-^ E14 embryos cultured for 10 days. Mean ± s.e.m of data collected from four separate experiments of *Cd40*^+/+^ neurons (control siRNA transfected, n = 27; PKC-β siRNA transfected, n = 26; PKC-γ siRNA transfected, n = 25) and *Cd40*^-/-^ neurons (control siRNA transfected, Fc treated, n = 34; control siRNA transfected, CD40-Fc treated, n = 26; PKC-γ siRNA transfected, Fc treated, n = 29; PKC-γ siRNA transfected, CD40-Fc treated, n = 24). **p<0.01 and ***p<0.001, actual p values provided in the text, one-way ANOVA with multiple Newman-Keuls tests.

DOI: https://doi.org/10.7554/eLife.30442.018

The following source data and figure supplement are available for figure 6:

**Source data 1.** Means, s.e.m. and n numbers for *Cd40*^+/+^ pyramidal neuron bar charts.
DOI: https://doi.org/10.7554/eLife.30442.020
**Source data 2.** Means, s.e.m. and n numbers for *Cd40*^+/+^ medium spiny neuron bar charts.
DOI: https://doi.org/10.7554/eLife.30442.021
**Source data 3.** Means, s.e.m. and n numbers for *Cd40*^-/-^ pyramidal neuron bar charts.
DOI: https://doi.org/10.7554/eLife.30442.022
**Source data 4.** Means, s.e.m. and n numbers for *Cd40*^-/-^ medium spiny neuron bar charts.
DOI: https://doi.org/10.7554/eLife.30442.023
**Figure supplement 1.** Expression of PKC isoforms in the hippocampus and striatum and siRNA knockdown of PKC-β and PKC-γ hippocampal and striatal cultures.
DOI: https://doi.org/10.7554/eLife.30442.019

branched than those of wild type littermates whereas the dendrite arbors of inhibitory striatal medium spiny neurons were greatly increased in size and more branched in the absence of CD40. These pronounced and opposite in vivo phenotypic changes were replicated in excitatory and inhibitory neurons cultured from *Cd40*^-/-^ mice. In both cases, these distinctive in vitro phenotypes were completely rescued in neurons cultured from *Cd40*^-/-^ mice by soluble CD40 (CD40-Fc), and were mimicked in neurons cultured from *Cd40*^+/+^ mice by soluble CD40L treatment. These observations together with our demonstration that both hippocampal excitatory and striatal inhibitory neurons display CD40L immunoreactivity, suggests that CD40 exerts its distinctive effects on dendrite arborisation in both excitatory and inhibitory neurons by a reverse signalling mechanism.

Our demonstration that both hippocampal excitatory pyramidal neurons and striatal inhibitory medium spiny neurons co-express CD40L and CD40 raise the possibility that CD40 activates CD40L in both kinds of neurons by an autocrine mechanism. We have previously reported that CD40-activated CD40L reverse signaling enhances axon growth from developing sympathetic neurons in vitro and contributes to the establishment of the sympathetic innervation of a specific subset of tissues in vivo (*McWilliams et al., 2015*). Because sympathetic neurons co-express CD40 and CD40L and can be cultured at exceptionally low density, it was possible to demonstrate that CD40-activates CD40L in sympathetic neurons by an autocrine mechanism. Because pyramidal neurons and MSNs have to be grown at much higher density and because these cultures contain a variety of cell types, including glial cells, it has not been possible to exclude the possibility that CD40L is also activated by paracrine mechanism in CNS neurons. Our demonstration that GFAP-positive astrocytes express CD40 and reports that microglia express low levels of CD40 (*Carson et al., 1998*; *Nguyen et al., 1998*) raise the possibility that CNS glia may play a role in activating CD40L in neurons. However, since CD40L and CD40 play a central role in immune responses (*Peters et al., 2009*), it is possible that microglial CD40 is related primarily to immune function, especially as CD40 is upregulated in these cells by proinflammatory cytokines (*Carson et al., 1998*; *Nguyen et al., 1998*).

A growing body of evidence has implicated several members of the TNFSF in the regulation of axon growth and tissue innervation in developing PNS by either forward or reverse signalling mechanisms. Typically, individual TNFSF members act on restricted populations of PNS neurons and either promote axon growth or inhibit axon growth-promoting effects of neurotrophins (*Desbarats et al., 2003*; *Gavaldà et al., 2009,2013*; *Kisiswa et al., 2013, 2017*; *McWilliams et al., 2015*; *O'Keeffe et al., 2008*; *Wheeler et al., 2014*). Here we not only demonstrate marked physiologically

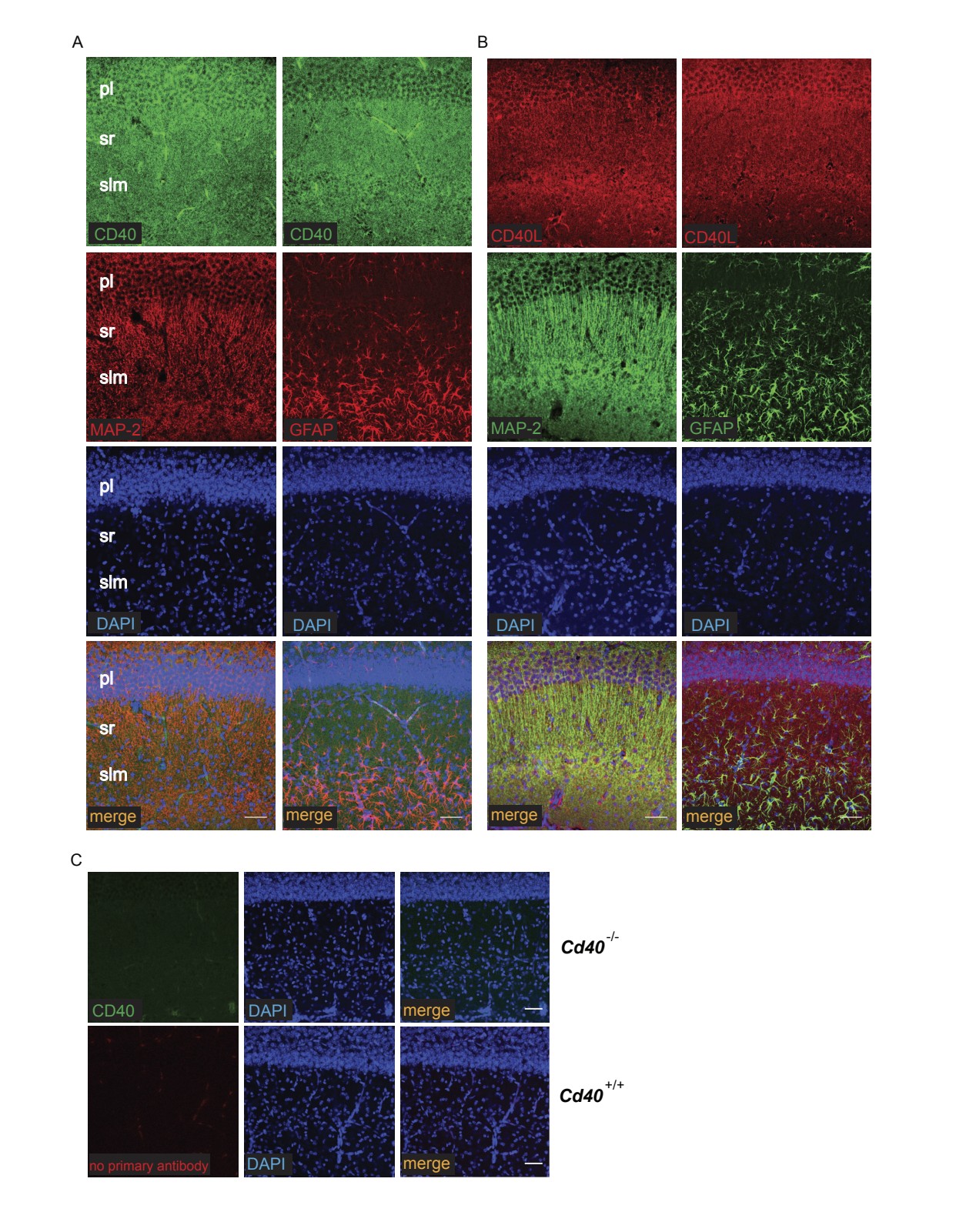

**Figure 7.** Localization of CD40 and CD40L in the developing hippocampus. Representative images of sections of the CA1 region of P9 hippocampi labelled with either anti-CD40 (**A**) or anti-CD40L (**B**) together with DAPI and either anti-MAP-2 or anti-GFAP (pl, pyramidal layer; sr, stratum radiatum; slm, stratum lacunosum moleculare). (**C**) Antibody specificity controls: sections of the hippocampus of P9 Cd40$^{-/-}$ mice incubated with the anti-CD40 antibody and sections of the hippocampus of P9 Cd40$^{+/+}$ mice that received no primary antibody. Scales bar, 50 µm.
*Figure 7 continued on next page*

*Figure 7 continued*

DOI: https://doi.org/10.7554/eLife.30442.024

The following figure supplement is available for figure 7:

**Figure supplement 1.** Expression of CD40 and CD40L in the developing hippocampus and striatum.
DOI: https://doi.org/10.7554/eLife.30442.025

relevant effects of CD40L reverse signalling on dendrite growth and branching in functionally different kinds of CNS neurons but show that CD40-activated CD40L reverse signalling can either promote or inhibit dendritic arborisation in different kinds of neurons. Our work reveals that a particular TNFSF member exerts distinctive, neuron-specific effects on CNS dendrite growth. In addition to the hippocampal and striatal neurons studied here, it is clear that CD40 and CD40L are widely expressed in the developing brain neurons (not shown). Thus, it is possible that CD40 has more widespread effects on the growth of neural processes in the developing brain than those documented here.

In addition to our demonstration that CD40 is physiologically relevant regulator of dendrite growth in the developing brain in vivo, we showed that CD40-activated CD40L-mediated reverse signalling promotes hippocampal pyramidal axon growth in vitro. This raises the possibility that CD40-activated CD40L-mediated reverse signalling may be a physiologically important regulator of pyramidal neuron axon growth in vivo. However, because the full extent of hippocampal pyramidal axons cannot be reliably discerned in Golgi preparations, we were not able to establish whether CD40L reverse signalling is physiological relevant for pyramidal axon growth in vivo. We have reported that the TNFSF member APRIL enhances the axon growth, but not dendrite growth, from cultured hippocampal pyramidal neurons (*Osório et al., 2014*), although the physiological relevance of this observation is unclear. However, we have recently shown that APRIL enhances axon growth from cultured midbrain dopaminergic neurons and that the initial nigrostriatal projection is significantly compromised in *April^-/-* embryos (*McWilliams et al., 2017*)

It has been reported that CD40 expression is retained in the adult brain, with most of the neurons of the dentate gyrus, hippocampal pyramidal layer and 60% of neurons in cerebral neocortex retaining expression (*Tan et al., 2002*). There is also some evidence for a neurodegenerative phenotype in aged CD40-deficient mice, with TUNEL-positive cells in the dentate gyrus, hippocampal pyramidal cell layer and neocortex and decreased level of neurofilament isoforms in whole brain lysates (*Tan et al., 2002*). Although the reason for these late degenerative changes is unclear, it is possible that the marked reduction in the size and complexity of pyramidal neuron dendritic arbors that we have documented in the postnatal and adult brain may contribute by impairing trophic maintenance of the degenerating neurons.

Our demonstration that pharmacological inhibition of PKC completely prevented the rescue of the distinctive dendrite phenotypes of CD40-deficient hippocampal and striatal neurons by soluble CD40-Fc and that pharmacological activation of PKC was as effective as CD40-Fc in rescuing these phenotypes suggests that PKC activation is an essential step in the effect of CD40-activated CD40L reverse signalling on dendrite growth in both kinds of neurons. This finding extends the importance of PKC in mediating the axon growth response of developing sympathetic neurons to TNFR1-activated TNF reverse signaling (*Kisiswa et al., 2017*) to another example of TNFSF reverse signaling affecting neural process growth. Moreover, it shows that PKC activation mediates both the growth-promoting and growth-inhibitory effects of reverse signaling in different kinds of neurons. PKC inhibition also prevents enhanced neurite growth from cultured embryonic hippocampal neurons by neuregulin-1 (*Gerecke et al., 2004*), whereas PKC activation increases dendrite growth and branching from these neurons by modulating neurotrophin synthesis (*Lim and Alkon, 2012*).

Using specific siRNA to knockdown the expression of different PKC isoforms, we have implicated the selective participation of PKC-β and PKC-γ in mediating the opposite effects of CD40L reverse signaling on dendrite growth from hippocampal pyramidal neurons and MSNs, respectively. While our findings do to exclude the participation of additional PKC isoforms in the signaling networks leading to increases or decreases in dendrite growth and branching, it will be intriguing in future work to ascertain how PKC-β and PKC-γ play these distinctive roles in neurons.

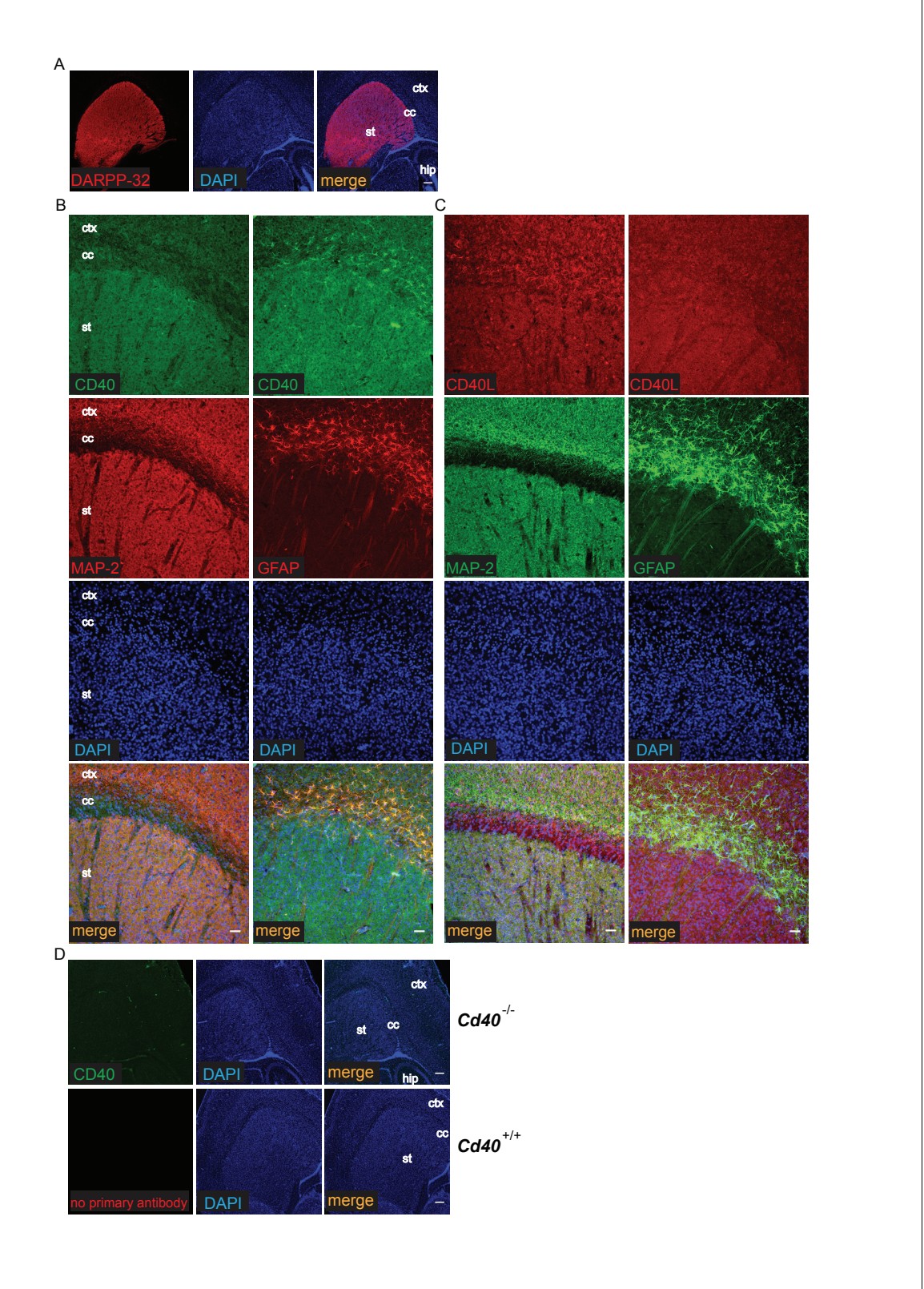

**Figure 8.** Localization of CD40 and CD40L in the developing striatum. (A) Low-power images of the expression of striatal MSNs labelled with DARPP-32 and double labelled with DAPI. Scale bar, 250 μm. Representative higher power images of sections of the striatum labelled with either anti-CD40 (B) or anti-CD40L (C) together with DAPI and either anti-MAP-2 or anti-GFAP. (D) Antibody specificity controls: sections of the striatum of P9 *Cd40*[-/-] mice

*Figure 8 continued on next page*

*Figure 8 continued*
incubated with the anti-CD40 antibody and sections of the striatum of P9 *Cd40*$^{+/+}$ mice that received no primary antibody (st, striatum; hip, hippocampus; ctx, cortex; cc, corpus callosum). Scale bar, 50 μm.
DOI: https://doi.org/10.7554/eLife.30442.026

Our current study extends the role of CD40L reverse signalling from modulating the growth of a subset of sympathetic axons in the PNS to that of a major, physiologically relevant regulator of dendrite growth and elaboration in the developing CNS, having distinctive and opposite effects on

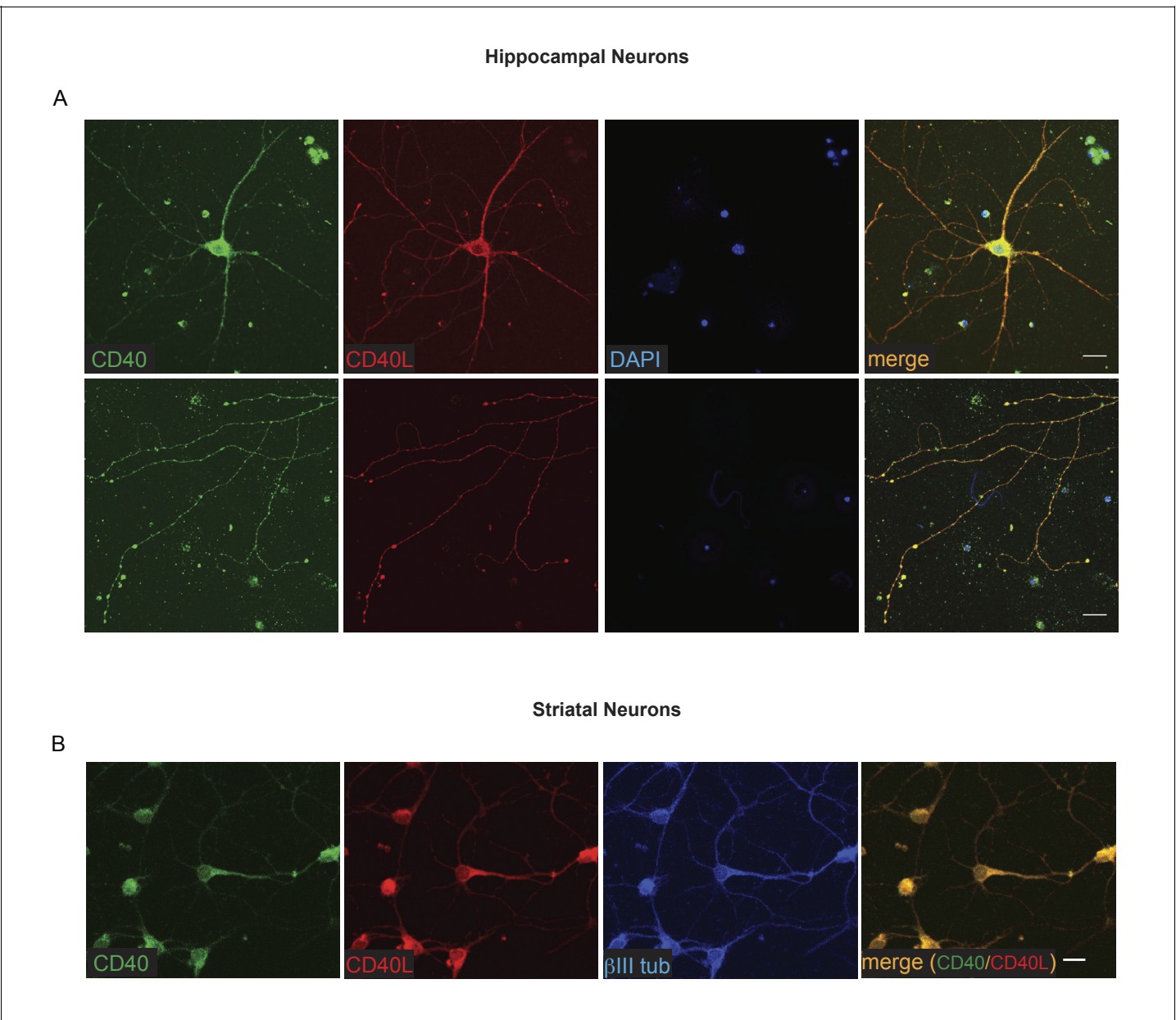

**Figure 9.** Co-localization of CD40 and CD40L in the cultured hippocampal pyramidal neurons and striatal MSNs. (**A**) Representative images of dissociated E18 hippocampal cultures after 9 days in vitro triple labelling with anti-CD40, anti-CD40L and DAPI. The upper panels show images of neuronal cell bodies and proximal neural processes. The lower panels show distal neural processes. (**B**) Representative images of dissociated E14 striatal cultures after 10 days in vitro triple labelling with anti-CD40, anti-CD40L and βIII tubulin. Scale bar, 20 μm.
DOI: https://doi.org/10.7554/eLife.30442.027

excitatory and inhibitory neurons. This work raises a host of important questions for future work, including how extensively CD40L reverse signalling regulates neuronal morphology and the functional properties of neural circuits, what are behavioural consequences of the major cellular phenotypes observed in CD40-deficient mice, its role in neural plasticity in the adult and what its contribution to the neurological sequelae of CNS inflammatory conditions might be.

## Materials and methods

### Mice

Mice were housed in a 12 hr light-dark cycle with access to food and water *ad libitum*. Breeding was approved by the Cardiff University Ethical Review Board and was performed within the guidelines of the Home Office Animals (Scientific Procedures) Act, 1986. *Cd40* null mutant mice in a C57BL6/J background were purchased from The Jackson Laboratory (Bar Harbor, Maine, USA). These mice were back-crossed for at least 10 generations into a CD1 background before undertaking any of the current experimental work. *Cd40*$^{+/-}$ mice were crossed to generate *Cd40*$^{+/+}$ and *Cd40*$^{-/-}$ littermates.

### Neuron culture

Primary hippocampal neuron cultures were prepared as described previously (*Kaech and Banker, 2006*), with modifications. Briefly, hippocampi were dissected from embryonic day 18 (E18) mouse fetuses and were triturated to produce a single cell suspension following trypsin digestion (Worthington, Lakewood, NJ, USA) and DNase I treatment (Roche Applied Science, UK). The neurons were plated in of 35 mm plastic dishes coated with poly-L-lysine (Sigma-Aldrich, UK) at a density of 15,000 cells/cm$^2$ and were cultured in Neurobasal A (Invitrogen, UK) supplemented with 2% B27 (Invitrogen, UK), 0.5 mM GlutaMAX I (Invitrogen, UK), 100 units/ml penicillin and 100 µg/ml streptomycin (Gibco BRL, UK).

Medium spiny neuron (MSN) cultures were prepared from E14 striatal primordia that were triturated to produce a single cell suspension following trypsin digestion (Worthington, Lakewood, NJ, USA) and DNase I treatment (Roche Applied Science, UK). Neurons were plated at a density of 15,000 cells/cm$^2$ at the center of 35 mm dishes coated with poly-L-lysine (Sigma-Aldrich, UK). Cells were cultured with Neurobasal A (Invitrogen, UK) supplemented with 2% B27 (Invitrogen, UK), 1% Foetal Calf Serum (FCS) (Sigma-Aldrich, UK), 100 units/ml penicillin and 100 µg/ml streptomycin (Gibco BRL, UK). To avoid extensive astrocyte proliferation, the medium was changed to medium without FCS after 7 days in vitro.

Hippocampal and striatal cultures were incubated at 37°C in a humidified atmosphere containing 5% CO$_2$ and in both cases were treated the day after plating. The culture medium was partially replaced with fresh medium with the treatments every 4–5 days. The cultures were treated with the following reagents as indicated in the text: soluble CD40L (sCD40L) (Enzo Life Sciences, Farmingdale, NY, USA, ALX-522–120 C010), CD40-Fc (Enzo Life Sciences, ALX-522–016 C050), Fc protein (Enzo Life Sciences, ALX-203–004 C050), Go6983 (Tocris Biosciences, UK, cat. no. 2285) and phorbol-12-myristate-13-acetate (PMA) (MERCK, UK, cat. no. 524400). Go6983 and PMA were reconstituted in DMSO and diluted in culture medium to the concentration indicated. Control cultures received an equivalent amount of DMSO. For PKC experiments, the cells were plated in 24 well dishes.

### Analysis of dendrite morphology

In hippocampal cultures, the neurite arbors of a proportion of the neurons were visualised in 3 day cultures by the fluorescent vital dye calcein-AM added at the end of the experiment and in 9 days cultures by transfecting the neurons with a GFP expression plasmid after 7 days in vitro using lipofectamine 2000 (Invitrogen, UK) according to the manufacturer's instructions with modifications. Briefly, the cultures were treated for 3 hr with a mixture of the expression vector and lipofectamine, after which they were washed with medium and cultured for a further 48 hr. The neurons were fixed for 30 min with 4% paraformaldehyde. MSN cultures were fixed with 4% paraformaldehyde after 10 days in vitro, after which they were permeabilized with 0.1% Triton-X100 and blocked with 0.5% BSA before labeling with anti-DARPP-32 (1:400 Cell Signaling Technology, Danvers, MA, USA) to identify MSNs. Analysis of neuronal morphology was studied after double labeling with anti-*β*III

tubulin (1:1500; chicken ab41489, AbCam, UK). After washing, the cultures were incubated with the appropriate fluorescent Alexa secondary antibodies (1:500; Thermofisher, UK, A-21206 and A-11042).

For the experiments of siRNA, 48 hr before fixing the neurons for the analysis of dendrite morphology, the neurons were co-transfected using lipofectamine with a GFP expression plasmid together with either of the following Silencer Select siRNA oligonucleotides at 10 nM: Silencer select negative control n.1, Prkcβ mouse and Prkcγ mouse (catalogue numbers 4390843, s71692 and s71693, respectively, Thermofisher, UK). For siRNA experiments on $Cd40^{-/-}$ neurons, the neurons were treated with either Fc or CD40-Fc after transfection.

Labelled neurons were visualized using a Zeiss LSM710 confocal microscope. Neurite arbors were assessed using Fiji (ImageJ) software with the semi-automated plugin Simple Neurite Tracer (*Longair et al., 2011*). All neurons labeled were counted without any criterion of exclusion. The mean and standard errors of the measurements from at least three independent experiments are plotted.

## Golgi preparations

Modified Golgi-Cox impregnation was performed on 150 μm coronal sections for the hippocampus and 100 μm parasagittal sections for the striatum of P10, P30 and adult mouse brains of $Cd40^{+/+}$ and $Cd40^{-/-}$ littermates using the FD Rapid GolgiStain kit (FD NeuroTechnologies, Ellicott City, MD, USA). Sholl analysis was carried out separately on the apical and basal dendritic arbors of pyramidal neurons in the CA1 hippocampal field and on the dendrite arbors of MSNs using the plugin Sholl Analysis of the Fiji software (*Schindelin et al., 2012*) after neuronal reconstruction with the plugin Simple Neurite Tracer (*Longair et al., 2011*).

## Immunoblotting

Hippocampal and striatal tissue was placed in triton lysis buffer supplemented with protease and phosphatase inhibitor cocktail mix (Sigma-Aldrich, UK). Cultured neurons were first washed with ice-cold PBS and lysed with triton lysis buffer supplemented with protease and phosphatase inhibitor cocktail mix. For experiments to confirm siRNA knock down, the neurons were grown at very high density in 60 mm tissue culture dishes.

Insoluble debris was removed by centrifugation at 14,000 g for 10 min at 4°C. Equal quantities of protein were run on SDS-PAGE gels, and were transferred to PVDF membranes (Immobilon-P, Millipore, UK) that were incubated with blocking solution (5% non-fat dry milk in TBS with 0.1% tween-20, TBS-T). After washing with TBS-T the blots were probed with the following antibodies: anti-CD40 (1:450; rat MAB4401, R and D, UK), anti-CD40L (1:700; rabbit ab2391, AbCam, UK), anti-PKC-a (1:1000, mouse 610107, BD Transduction Laboratories, UK), anti-PKC-b (1:250, mouse 610127, BD Transduction Laboratories, UK), anti-PKC-γ (1:1000; rabbit 43806S, Cell Signaling Technology, Danvers, MA, USA), anti-PKC-ε (1:1000, mouse 610085, BD Transduction Laboratories, UK), anti-GAPDH (1:90,000; mouse G8795, Sigma, UK) or anti-actin (1;60,000: mouse ab3280, AbCam, UK). Bound primary antibodies were visualized with HRP-conjugated goat anti-rat (1:1000, 7077, Cell Signaling Technology, Danvers, MA, USA), donkey anti-rabbit or anti-mouse secondary antibodies (1:5000; rabbit W4011, mouse W4021, Promega, UK) and EZ-ECL kit Enhanced Chemiluminescence Detection Kit (Biological Industries, Geneflow Limited, UK).

## Immunohistochemistry and immunocytochemistry

For immunohistochemistry, P9 brains were fixed in fresh 4% paraformaldehyde in 0.12 M phosphate buffer, pH 7.2 for 24 hr at 4°C. After washing in PBS, the tissue was cryoprotected in 30% sucrose before being frozen. Tissue was frozen in isopentene cooled with dry ice and was serially sectioned at 30 μm. After washing with PBS, the sections were permeabilized with 0.1% Triton X-100 (Sigma-Aldrich, UK) for 1 hr and then blocked with 1% BSA and 0.1% Triton X-100 (Sigma-Aldrich, UK) in PBS for 2 hr at room temperature. Primary antibodies were prepared in PBS with 0.5% BSA and 0.1% Triton X-100, and incubated overnight at 4°C. Primary antibodies were: rabbit polyclonal anti-CD40L (1:150, AbCam, UK, catalogue number ab2391), rat monoclonal anti-CD40 (1:100, AbCam, UK, catalogue number ab22469), chicken anti-MAP2 (1:5000, AbCam, UK, catalogue number ab5392), chicken anti-GFAP (1:2000, AbCam, UK, catalogue number ab4674), rabbit monoclonal

anti-DARPP-32 (1:400 Cell Signaling Technology, Danvers, MA, USA, catalogue number 2306) and chicken anti-$\beta$III tubulin (1:1500, AbCam, UK, catalogue number ab41489). The sections were washed in PBS and were incubated for 1 hr with secondary fluorescent Alexa antibodies (1:500; Thermofisher, UK, A-21207; A-11006; A-11042; A-11039; AbCam, ab175674). Negative controls (no primary antibody and staining of tissues from $Cd40^{-/-}$ mice) were also set up. After washing with PBS, the sections were mounted on slices with Fluoromount-G (Cambridge Bioscience, UK). Images were obtained with Zeiss LSM710 confocal microscope.

For immunocytochemistry, cultures were fixed as above for 15 min at room temperature and were washed extensively with PBS before permeabilization and blocking of nonspecific binding for 1 hr at room temperature as above. The cultures were incubated overnight at 4°C with rabbit polyclonal anti-CD40L (1/200), rat monoclonal anti-CD40 (1/200) and for the striatal cultures also with anti-$\beta$III tubulin (1/1500). After washing with PBS, the cultures were incubated with donkey anti-rabbit Alexa 594, goat anti-rat Alexa 488 and goat anti-chicken Alexa 405 (1:500, Thermofisher, UK; A-21207; A-11006; AbCam, ab175674). Images were obtained using a Zeiss LSM710 confocal microscope.

## Acknowledgements

This work was supported by the Wellcome Trust (grant number 103852). Our thanks to Dr Suppalak Lewis for help in optimising the immunohistochemistry conditions.

## Additional information

### Funding

| Funder | Grant reference number | Author |
| --- | --- | --- |
| Wellcome Trust | 103852 | Alun M Davies |

The funders had no role in study design, data collection and interpretation, or the decision to submit the work for publication.

### Author contributions

Paulina Carriba, Conceptualization, Formal analysis, Investigation, Methodology, Writing—review and editing; Alun M Davies, Conceptualization, Formal analysis, Supervision, Funding acquisition, Writing—original draft, Project administration, Writing—review and editing

### Author ORCIDs

Paulina Carriba (iD) http://orcid.org/0000-0002-6980-2277
Alun M Davies (iD) http://orcid.org/0000-0001-5841-8176

### Ethics

Animal experimentation: Mouse breeding and housing was approved by the Cardiff University Ethical Review Board and was performed within the remit of Home Office Licence PPL303081.

### Decision letter and Author response

Decision letter https://doi.org/10.7554/eLife.30442.029
Author response https://doi.org/10.7554/eLife.30442.030

## Additional files

### Supplementary files

• Transparent reporting form
DOI: https://doi.org/10.7554/eLife.30442.028

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
