## [Decision Letter]

Thank you for submitting your article "CD40 is a major regulator of dendrite growth from developing excitatory and inhibitory neurons" for consideration by *eLife*. Your article has been reviewed by three peer reviewers, and the evaluation has been overseen by a Reviewing Editor and Marianne Bronner as the Senior Editor. The following individual involved in review of your submission has agreed to reveal her identity: Bonnie Firestein (Reviewer #1).

Three expert reviewers have assessed the manuscript and their views, as well as my own, form the basis of this letter.

We were all impressed with the novelty and significance of your findings with CD40 and CD40L and with the combination of in vitro and in vivo approaches. The different dendritic responses of different neuronal types is especially intriguing.

We would like to invite you to submit a revised manuscript, taking into account the suggestions of the three reviewers (appended here). The most ambitious suggestion (relating to a behavioral phenotype; point 2 from reviewer 2) is likely outside of the scope of your current body of work.

*Reviewer #1:*

This manuscript reports a completely novel finding on the role of CD40 and CD40L on shaping the dendritic arbor. The work uses both in vivo and in vitro analysis and identifies PKC as a mediator of CD40 function. Since CD40 and CD40L have generally been studied in the immune system, this work will have a very high impact on both fields. Given that the authors uncover distinct effects on excitatory and inhibitory neurons, this work is distinct from the work on CD40 and CD40L in the PNS.

The manuscript is well-written and very easy to follow. The data are of high quality and appropriate controls were performed.

I have two questions.

1) Did the authors observe any changes to inhibitory neurons in the hippocampus in the knockout animals? Or in culture (although this may be difficult to study as only 10% of neurons are inhibitory)?

2) Please perform two-way ANOVA for Sholl curve statistics.

In sum, this is a complete story.

*Reviewer #2:*

Carriba and Davies report on a novel role of CD40/CD40 ligand signaling in developing CNS neurons. Interestingly, the effects of CD40 gene knockout on hippocampal pyramidal neurons and striatal medium spiny neurons (MSNs) are opposite, both in vivo and in vitro: Loss of CD40 decreased hippocampal neuron complexity, whereas MSN complexity increased. The authors then used soluble CD40L as dominantly-interfering agent to inhibit CD40/CD40L interaction in cultured wild-type neurons, and soluble CD40-Fc to activate CD40L reverse signaling in cultured CD40-/- neurons. The results suggest that CD40-induced CD40L reverse signaling promotes and restricts growth of hippocampal neurons and MSNs, respectively. In a first attempt to get at the underlying mechanism, the authors pharmacologically interfered with PKC activity and found, somewhat unsatisfactorily, that PKC activity mediates both the growth of hippocampal neurons and growth suppression of MSNs.

I find that the experiments were executed with care and competence, the results convincing and reasonably comprehensive, and most importantly unexpected and somewhat surprising, such that they may spark the interest of readers outside the immediate field of TNF superfamily cytokines.

1) The in vivo neuron complexity data were taken from only one time point (P10). To raise significance of their findings, the authors need to add at least one later time point (e.g. P30) to exclude that the effects of CD40 elimination on hippocampal neurons and MSNs are transient.

2) The observation of a morphological change in a mutant mouse brain is interesting and an excellent starting point; however, the description of a correlating alteration in physiological properties of the neurons or in a relevant behavioral paradigm would be so much more interesting and at the same time validate the observed morphological changes.

*Reviewer #3:*

The manuscript by Carriba and Davies examines the role of CD40, an integral membrane protein, in dendritic development. Several studies have demonstrated that activation of CD40 by its ligand CD40L or CD154 in T cells results in the activation of antigen presenting cells. However, the role of these proteins remains poorly understood in the nervous system.

By analysing CD40 knock out mutant mice the authors discovered the exciting find that this protein plays contrasting roles in dendritic arborisation in different neuronal cell types. Loss of function of CD40 in hippocampal neurons results in decreased dendritic branching. In contrast, in medium spiny neurons (MSNs) in the striatum deficiency in CD40 results in the formation of more elaborated dendrites. Intriguingly, the suppression or promotion of dendritic development is mediated through the ligand CD40L. Moreover, using pharmacological approaches, the authors demonstrated a role of PKC in mediating these effects.

The paper presents new and exciting data. Importantly, the experiments are well conducted and the phenotype is striking and clear. However, the authors need to examine in more detail the mechanisms by which CD40 promotes and suppresses dendritic arborisation in these different cell types. In summary, this paper reports a very exciting finding worth publishing in *eLife*. However, the authors should examine in more detail the mechanisms involved to begin to elucidate how CD40 elicits such contrasting effects.

1) Given that expression of CD40 in the CNS has not been well reported, the authors should explain why they focused their work on the hippocampus and striatum to examine the phenotype of the CD40 mutant mice.

2) The data presented in all the figures is clear, convincing and well quantified. However, it is not clear why the neurons in Figure 2 top panels (CD40+/+ and CD40-/-) are different from those represented in Figure 5 (DMSO and Fc). The neurons were isolated from the same stage and cultured for the same period of time.

3) In Figure 2, the authors present data demonstrating a role for PKC in mediating the effects of CD40 using pharmacological approaches. It would be important to also present complementary results using for example genetic approaches to validate the role of PKC.

4) Figure 6—figure supplement 1, the authors clearly demonstrate the specificity of the CD4 by western blots. A similar control using the brain slices should be used to corroborate the staining in Figure 6.

5) The authors indicated in the first paragraph of the Discussion that they demonstrated that both hippocampal and striatal neurons "display surface CD40L immunoreactivity". However, according to the Materials and methods section, the staining was done after treating the cells with detergents. Thus, the staining reflects the total levels of the protein but not just surface. This statement should be changed.

6) Regarding the mechanisms that could modulate the opposing effects of CD40. One possibility is that the authors explore the precise location of the CD40 and CD40L along the dendrites and whether Rho GTPases are differentially regulated in different neurons.

---

## [Author Response]

Reviewer #1:[…] I have two questions.1) Did the authors observe any changes to inhibitory neurons in the hippocampus in the knockout animals? Or in culture (although this may be difficult to study as only 10% of neurons are inhibitory)?

As pointed out by the reviewer, inhibitory neurons comprise only very small proportion of neurons in the hippocampus. What’s more, there are many kinds of inhibitory neurons with different morphologies in the hippocampus, which further complicates addressing this question. For these and other reasons, we focused on the large population of well-characterized inhibitory neurons of the striatum that we have been able to study extensively in vivo and in vitro. While it is indeed an interesting question whether CD40 signaling affects dendrite growth from different kinds of inhibitory neurons, this would require very extensive work that is beyond the scope of our initial study.

2) Please perform two-way ANOVA for Sholl curve statistics.

Done.

In sum, this is a complete story.Reviewer #2:[…] 1) The in vivo neuron complexity data were taken from only one time point (P10). To raise significance of their findings, the authors need to add at least one later time point (e.g. P30) to exclude that the effects of CD40 elimination on hippocampal neurons and MSNs are transient.

We have studied the hippocampus and striatum of CD40-deficient mice and wild type littermates at two additional ages (P30 and adult) by the Golgi technique. We find very clear differences at both ages, similar to those observed in the developing hippocampus and striatum: the dendrites of hippocampal pyramidal were markedly stunted in CD40-deficient mice, whereas those striatal inhibitory neurons were much more exuberant. Representative images at these older stages are shown in Figure 1—figure supplement 2.

2) The observation of a morphological change in a mutant mouse brain is interesting and an excellent starting point; however, the description of a correlating alteration in physiological properties of the neurons or in a relevant behavioral paradigm would be so much more interesting and at the same time validate the observed morphological changes.

How the distinctive and widespread morphological changes observed in dendritic arbors of CD40-deficient mice affects behaviour is indeed a fascinating and important question. However, we agree with Editor’s comment “that this likely is outside the scope of our current body of work”. We are not a behavioral laboratory and have no expertise in this field. However, we have no doubt that our observations will be of great interest to behavioral laboratories and are likely to initiate studies to ascertain in which ways behavior is affected.

Reviewer #3:The manuscript by Carriba and Davies examines the role of CD40, an integral membrane protein, in dendritic development. Several studies have demonstrated that activation of CD40 by its ligand CD40L or CD154 in T cells results in the activation of antigen presenting cells. However, the role of these proteins remains poorly understood in the nervous system.By analysing CD40 knock out mutant mice the authors discovered the exciting find that this protein plays contrasting roles in dendritic arborisation in different neuronal cell types. Loss of function of CD40 in hippocampal neurons results in decreased dendritic branching. In contrast, in medium spiny neurons (MSNs) in the striatum deficiency in CD40 results in the formation of more elaborated dendrites. Intriguingly, the suppression or promotion of dendritic development is mediated through the ligand CD40L. Moreover, using pharmacological approaches, the authors demonstrated a role of PKC in mediating these effects.The paper presents new and exciting data. Importantly, the experiments are well conducted and the phenotype is striking and clear. However, the authors need to examine in more detail the mechanisms by which CD40 promotes and suppresses dendritic arborisation in these different cell types. In summary, this paper reports a very exciting finding worth publishing in eLife. However, the authors should examine in more detail the mechanisms involved to begin to elucidate how CD40 elicits such contrasting effects.1) Given that expression of CD40 in the CNS has not been well reported, the authors should explain why they focused their work on the hippocampus and striatum to examine the phenotype of the CD40 mutant mice.

There are numerous reasons why we focused on the hippocampus and striatum in our initial study. Hippocampal pyramidal and striatal medium spiny neurons are among the most extensively characterized major classes of neurons in the developing CNS. They have characteristic morphologies in vivo, facilitating studies of dendrite morphology in the intact animal. They can be easily cultured as the predominant cell type in cultures set up from particular embryonic stages, facilitating signaling studies. Because the stages of dendritogensis are very well-characterized in these neurons in vitro, cultures are especially suitable for studying the factors that influence dendrite growth and branching. We make these points in the revised text.

CD40 and CD40L are indeed widely expressed in CNS, and we have preliminary evidence that CD40 reverse signaling influences the growth of dendrites from cortical pyramidal neurons, but have not followed this up because dendritogenesis in hippocampal pyramidal neurons is a much better model system in which to study this.

2) The data presented in all the figures is clear, convincing and well quantified. However, it is not clear why the neurons in Figure 2 top panels (CD40+/+ and CD40-/-) are different from those represented in Figure 5 (DMSO and Fc). The neurons were isolated from the same stage and cultured for the same period of time.

There are two issues here. First, to match the size of the micrographs with the size of the bar charts in Figure 2 and Figure 5, we used different magnifications, making the scale bars different lengths. This gives the impression that the neurons shown in Figure 2 are larger than the equivalent neurons shown in Figure 5. To avoid confusion, we have adjusted the magnification so the scale bars in these figures are the same length in both figures. Second, there is a good deal of variation in the morphology of individual neurons, making selection of “representative neurons” problematic. For this reason, we quantify a very large number of neurons in multiple experiments. So what’s really important is the quantification shown in the bar charts. Indeed, this is remarkably consistent given that the different kinds of experiments shown in Figure 2 and Figure 5 were done over 3 years. We show micrographs only to give the reader some idea of what the cultured neurons look like. That said, we have selected a few different micrographs where necessary.

3) In Figure 2, the authors present data demonstrating a role for PKC in mediating the effects of CD40 using pharmacological approaches. It would be important to also present complementary results using for example genetic approaches to validate the role of PKC.

To address this issue, we studied the effects of siRNA knock down of selected PKC isoforms on dendrite growth from hippocampal pyramidal neurons and MSNs. Because PMA activates conventional and novel PKC isoforms, we first checked expression of the conventional and novel PKC isoforms that have been reported to be abundantly expressed in whole brain. Using Western blotting, we demonstrated expression of PKC-α, β, γ and ε in the developing hippocampus and striatum. Because we were able to consistently and efficiently knock down PKC-β and PKC-γ expression in hippocampal and striatal neuron cultures with specific siRNA, we focused subsequent experiments on these isoforms. We were able to show that these isoforms selectively participate in mediating the opposite effects of CD40L reverse signaling on dendrite growth from hippocampal and striatal neurons.

In initial experiments using wild type neurons, we showed that transfection of PKC-β siRNA but not PKC-γ siRNA significantly decreased dendrite growth from hippocampal neurons and that transfection of PKC-γ siRNA but not PKC-β siRNA significantly increased dendrite growth from MSNs. In subsequent experiments on CD40-deficient neurons, we showed that PKC-β siRNA completely prevented CD40-Fc from rescuing the stunted dendrite phenotype of CD40-deficient hippocampal pyramidal neurons and that PKC-γ siRNA significantly prevented CD40-Fc from rescuing the exuberant dendrite phenotype of CD40-deficient MSNs.

In summary, we have not only used an alternative experimental approach to validate the role of PKC, but have additionally shown that different PKC isoforms are selectively involved in hippocampal and striatal neurons. These data are reported in new Figure 6 and Figure 6—figure supplement 1.

4) Figure 6—figure supplement 1, the authors clearly demonstrate the specificity of the CD4 by western blots. A similar control using the brain slices should be used to corroborate the staining in Figure 6.

We include these specificity controls for sections of the hippocampus and striatum (Figure 7 and Figure 8).

5) The authors indicated in the first paragraph of the Discussion that they demonstrated that both hippocampal and striatal neurons "display surface CD40L immunoreactivity". However, according to the Materials and methods section, the staining was done after treating the cells with detergents. Thus, the staining reflects the total levels of the protein but not just surface. This statement should be changed.

This oversight has been corrected in the text.

6) Regarding the mechanisms that could modulate the opposing effects of CD40. One possibility is that the authors explore the precise location of the CD40 and CD40L along the dendrites and whether Rho GTPases are differentially regulated in different neurons.

To begin to explore this intriguing possibility, we used Western blotting to assess the relative levels of expression of RhoA/B/C, Rac1/2/3 and Cdc42 in developing hippocampus and striatum. While all proteins were expressed in these tissues, there were no obvious differences in the relative levels between these tissues. We used immunohistochemistry and immunocytochemistry in wild type and CD40-deficient neurons to ascertain whether there were differences in the distribution of these proteins, but we did not observe any obvious differences between hippocampal and striatal neurons. Finally, considering that there is a redistribution of these proteins when they are activated, we also tried to detect changes in the distribution after treating CD40-deficient neurons with CD40-Fc, but again, we did not observe any clear differences in distribution. Although these results do not exclude that possibility that Rho GTPases play a role in the differential response of pyramidal and medium spiny neurons to CD40 reverse signaling, these results do not give us sufficient reason to pursue this line of enquiry at this stage.